


# Effects of climatic seasonality on the isotopic composition of evaporating soil waters

Paolo Benettin[1], Till H. M. Volkmann[2], Jana von Freyberg[3,4], Jay Frentress[5], Daniele Penna[6], Todd E. Dawson[7], and James W. Kirchner[3,4,8]

[1]Laboratory of Ecohydrology ENAC/IIE/ECHO, École Polytechinque Fédérale de Lausanne (EPFL), Lausanne, Switzerland
[2]Biosphere 2, University of Arizona, Tucson, AZ USA
[3]Department of Environmental Systems Science, ETH Zurich, Zürich, Switzerland
[4]Research Unit Mountain Hydrology and Mass Movements, Swiss Federal Institute for Forest, Snow and Landscape Research (WSL), Birmensdorf, Switzerland
[5]Faculty of Science and Technology, Free University of Bolzano, Italy
[6]Department of Agricultural, Food and Forestry Systems, University of Florence, Italy
[7]Department of Integrative Biology, University of California, Berkeley, CA USA
[8]Department of Earth and Planetary Science, University of California, Berkeley, CA USA

*Correspondence to:* Paolo Benettin (paolo.benettin@epfl.ch)

**Abstract.** Stable water isotopes are widely used in ecohydrology as tracers of the transport, storage, and mixing of water on its journey through landscapes and ecosystems. Evaporation leaves a characteristic signature on the isotopic composition of the water that is left behind, such that in dual-isotope space, evaporated waters plot below the Local Meteoric Water Line (LMWL) that characterizes precipitation. Soil and xylem water samples can often plot below the LMWL as well, suggesting that they have also been influenced by evaporation. These soil and xylem water samples frequently plot along linear trends

in dual-isotope space. These trendlines are sometimes termed "evaporation lines" and their intersection with the LMWL is sometimes interpreted as the isotopic composition of the precipitation source water. Here we use numerical experiments based on established isotope fractionation theory to show that these trendlines are often by-products of the seasonality in evaporative fractionation and in the isotopic composition of precipitation. Thus, they are often not true evaporation lines, and, if interpreted

as such, can yield highly biased estimates of the isotopic composition of the source water.

*Copyright statement.* TEXT

# 1   Introduction

Stable water isotopes ([18]O and [2]H) are widely used in ecohydrology as tracers of the transport, storage, and mixing of water, from its origin as precipitation, through the soil, and ultimately to groundwater and streamflow (Kendall and McDonnell,

1998) or to plant uptake and transpiration (Dawson and Ehleringer, 1998). Water isotopes also reflect evaporation losses, through the progressive enrichment of [18]O and [2]H in the remaining liquid. Past applications of stable water isotopes in soil hydrology studies have included identifying evaporation fronts in the unsaturated zone (e.g. Allison and Barnes, 1983; Dawson





and Ehleringer, 1998; Rothfuss et al., 2015; Sprenger et al., 2017), quantifying groundwater recharge rates and mechanisms (e.g. Healy and Scanlon, 2010; McGuire and McDonnell, 2006; Sakakibara et al., 2017), quantifying root water uptake (e.g. Ehleringer and Dawson, 1992; Dawson et al., 2002; Volkmann et al., 2016; Rothfuss and Javaux, 2017), and identifying plant water sources (e.g. Brooks et al., 2010; Dawson and Ehleringer, 1991; Dawson and Simonin, 2011; Goldsmith et al.,

2012; Evaristo et al., 2015; Hervé-Fernández et al., 2016; McCutcheon et al., 2017). A recent review by Sprenger et al. (2016) provides an extensive overview of isotope-based studies in the unsaturated zone. When expressed in the conventional $\delta$ notation and displayed together in a so-called dual-isotope plot (e.g., Figure 1), $\delta^{18}O$ and $\delta^2 H$ in precipitation at any given location will typically follow a linear trend (Craig, 1961; Dansgaard, 1964) termed the Local Meteoric Water Line (LMWL). As a water parcel evaporates, its isotopic composition will evolve along an evaporation line whose slope is determined by the relative

evaporation rates of the different water isotopologues (Figure 1). This evaporation line will generally have a shallower slope than the LMWL.

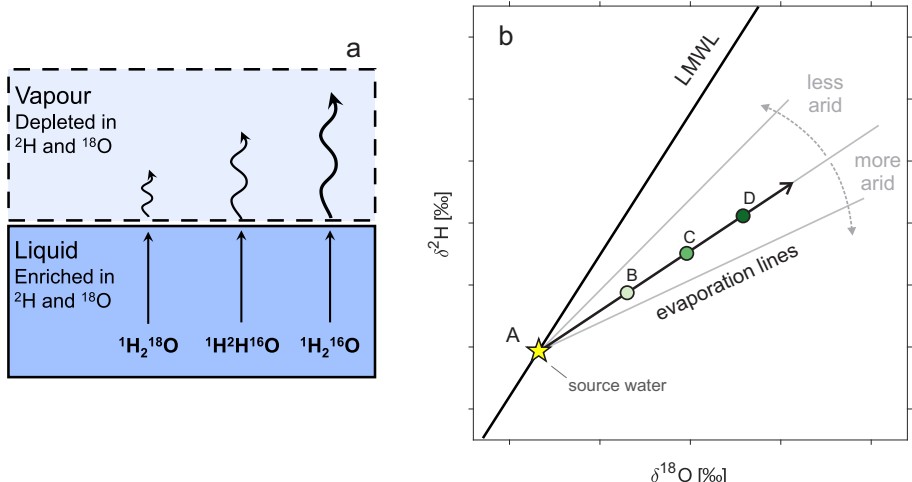

**Figure 1.** Fractionation effects during evaporation from an open water body. (a): Heavier water molecules ($^1H^2H^{16}O$, $^1H_2^{18}O$) break their bonds and evaporate less readily than lighter water molecules ($^1H_2^{16}O$), and thus have lower saturation vapour pressures. Heavier molecules also diffuse away from the evaporating surface less rapidly. As a consequence, during evaporation, lighter water molecules vaporize faster than heavier water molecules. The ratios between the evaporation rates of the different water isotopologues (net of any condensation) determine the slope of the evaporation line describing the progressive isotopic enrichment of the liquid water that is left behind. (b): Progressive enrichment (dots B-D) of a water source (yellow star A). The evaporation line typically lies below the local meteoric water line (LMWL), at an angle that depends on the aridity and the isotopic composition of the atmosphere (and thus on the relative rates of re-condensation of each isotopologue). Panel (a) was adapted from Leibundgut et al. (2009).

   Collections of soil and xylem water samples also often lie at an angle to the LMWL, and are often well described by linear fits (e.g. Brooks et al., 2010; Dawson and Simonin, 2011; Goldsmith et al., 2012; Evaristo et al., 2015). If these lines are evaporation lines, then extrapolating them to their intersection with the LMWL should yield the original composition of the

pre-evaporation source water. Exactly this strategy has been used to infer source water compositions for soil water and xylem





water (e.g. Evaristo et al., 2015; Hervé-Fernández et al., 2016; Javaux et al., 2016), as well as groundwaters (e.g. Dogramaci et al., 2012) and streamwaters (e.g. Telmer and Veizer, 2000). Information on the source water composition is then typically used to draw conclusions about water cycling processes in the study systems. This inference should be valid if the evaporated samples all originate from a single source water. But what if they don't? Is a linear trend, by itself, sufficient evidence that the

trend is actually an evaporation line? To date, no benchmark experiment has tested whether, and under what conditions, the trendline passing through fractionated soil water samples correctly identifies their source water.

    Here we use simple numerical experiments, based on established isotope fractionation theory, to model the isotopic evolution of seasonally-varying precipitation inputs, under the influence of seasonally-varying evaporation processes. These simulations show that the resulting evaporated samples often fall along well-defined linear trends that are markedly different from evapo-

ration lines, and therefore do not point to any meaningful source water composition.

## 2   Materials and Methods

We simulate the isotopic composition of evaporating soil waters using equations based on the simple and widely used linear resistance model of Craig and Gordon (1965). We then introduce the effect of climatic seasonality by applying these equations to seasonally varying isotopic sources and atmospheric conditions.

**2.1   Evaporative fractionation in soils**

The Craig and Gordon (1965) model estimates the joint effect of equilibrium and kinetic isotopic fractionation during the phase transition from liquid water to vapour. When resistance to transport in the liquid phase is neglected, the isotopic composition of the water vapour flux can be expressed as:

$$\delta_E = \frac{(\delta_L - \varepsilon^+)/\alpha^+ - h \cdot \delta_A - \varepsilon_k}{1 - h + 10^{-3} \cdot \varepsilon_k} \tag{1}$$

where $\delta_L$ and $\delta_A$ indicate the isotopic compositions of the evaporating surface and the atmosphere, $h$ is the relative humidity of the atmosphere, $\alpha^+$ and $\varepsilon^+$ are equilibrium fractionation factors, and $\varepsilon_k$ is a kinetic fractionation factor. Here and in the following, $\delta$ values and fractionation factors may refer to either hydrogen or oxygen isotopes unless otherwise noted. The $\delta$ notation expresses water isotope ratios as deviations, in parts per thousand, from Vienna Standard Mean Ocean Water (Kendall and Caldwell, 1998).

The equilibrium fractionation factor $\alpha^+$ [-] describes differences between the isotopic compositions of liquid and vapour phases at isotopic equilibrium. It is expressed here as the super-ratio of liquid to vapour isotope ratios, and its value is slightly larger than one, reflecting the fact that lighter molecules break their bonds more readily and thus are more abundant in the vapour phase. The values of $\alpha^+$ can be computed as a function of temperature $T$ [K] using the well-established experimental





results by Horita and Wesolowski (1994):

$$10^3 \ln[\alpha^+(^2\mathrm{H})] = 1158.8(T^3/10^9) - 1620.1(T^2/10^6) + 794.84(T/10^3) \tag{2}$$
$$- 161.04 + 2.9992(10^9/T^3)$$

$$10^3 \ln[\alpha^+(^{18}\mathrm{O})] = -7.685 + 6.7123(10^3/T - 1.6664(10^6/T^2) + 0.3504(10^9/T^3) \tag{3}$$

The equilibrium isotopic separation between liquid and vapour is then computed as $\varepsilon^+ = (\alpha^+ - 1) \, 10^3$ [‰].

The kinetic fractionation factor $\varepsilon_k$ quantifies isotopic effects during net evaporation associated with the higher diffusivities of isotopically lighter molecules. Variations in $\varepsilon_k$ are generally dominated by the relative humidity ($h$) of the air overlying the evaporating surface. Several expressions have been derived specifically for $\varepsilon_k$ in soils (see Mathieu and Bariac, 1996; Soderberg et al., 2012). Here we use a simplified expression given as (Gat, 1996; Horita et al., 2008):

$$\varepsilon_k = \theta \, n \, (1 - h) \, (1 - D_i/D) \, 10^3 \quad [‰] \tag{4}$$

The weighting term $\theta$ [-] accounts for the possible influence of the evaporation flux on the ambient moisture, and is usually assumed to equal 1 for small water bodies (Gat, 1996). The term $D_i/D$ is the ratio between the diffusivities of the heavy and light isotopes. Commonly accepted values are provided by Merlivat (1978): $D_i/D(^2\mathrm{H}) = 0.9755$ and $D_i/D(^{18}\mathrm{O}) = 0.9723$. The term $n$ [-] accounts for the aerodynamic regime above the evaporating liquid-vapour interface. It ranges from $n = 0.5$ (fully turbulent transport that reduces kinetic fractionation, appropriate for lakes or saturated soil conditions) to $n = 1$ (fully diffusive transport, appropriate under very dry soil conditions). According to equation (4), in a dry atmosphere ($h = 0$), the kinetic fractionation factor is roughly 12.2-24.5 ‰ for $\varepsilon_k(^2\mathrm{H})$ and 13.8-27.7 ‰ for $\varepsilon_k(^{18}\mathrm{O})$.

We now consider the case of an isolated volume of water with initial isotopic composition $\delta_0$ that evaporates into the atmosphere. As evaporation is the only flux, the water volume decreases in time (a case sometimes referred to as a "desiccating" water body). We use $x$ [-] to represent the fraction of the initial volume that has evaporated. The fraction remaining as liquid thus equals $1 - x$. Assuming that the fractionation factors do not change during the evaporation process, the equation describing the isotopic composition of the residual liquid $\delta_L$ is (Gonfiantini, 1986):

$$\delta_L = (\delta_0 - \delta^*)(1 - x)^m + \delta^* \tag{5}$$

where $\delta^*$ [‰] represents the limiting isotopic composition (i.e. the composition that the desiccating water volume would approach upon drying up) and the term $m$ [-] is referred to as "temporal enrichment slope" (Gibson et al., 2016). These two terms can be computed as:

$$\delta^* = (h\delta_A + \varepsilon_k + \varepsilon^+/\alpha^+)/(h - 10^{-3} \cdot (\varepsilon_k + \varepsilon^+/\alpha^+)) \tag{6}$$

and

$$m = (h - 10^{-3} \cdot (\varepsilon_k + \varepsilon^+/\alpha^+))/(1 - h + 10^{-3} \cdot \varepsilon_k) \tag{7}$$





Equation (5) can represent an isolated volume of precipitation with initial isotopic composition $\delta_P$ that progressively evaporates into an atmosphere with isotopic composition $\delta_A$. If the isotopic composition of the atmospheric vapour is unknown, it is common to assume that it is in equilibrium with precipitation (Gibson et al., 2008):

$$\delta_A = (\delta_P - 10^{-3} \cdot \varepsilon^+)/\alpha^+ \tag{8}$$

As an introductory example, we modeled the isotopic evolution of an individual water volume by implementing equations (1-8) with parameters $T = 20$ [°C], $h = 0.75$ and $n = 1$. Figure 2 illustrates the resulting increase in the hydrogen and oxygen $\delta_L$ of the residual water during the evaporation process. The dual-isotope plot (Figure 2c) shows the simultaneous behavior of both hydrogen and oxygen isotope ratios. As more of the water evaporates, the composition of the residual liquid gradually departs from the LMWL following a nearly linear trajectory. This trajectory is termed the evaporation line. Depending on the atmospheric parameters used in equations (1-8), the slopes of evaporation lines will typically range from 2.5 to 5, markedly shallower than typical meteoric water lines, which usually have slopes of roughly 8 (Kendall and Caldwell, 1998).

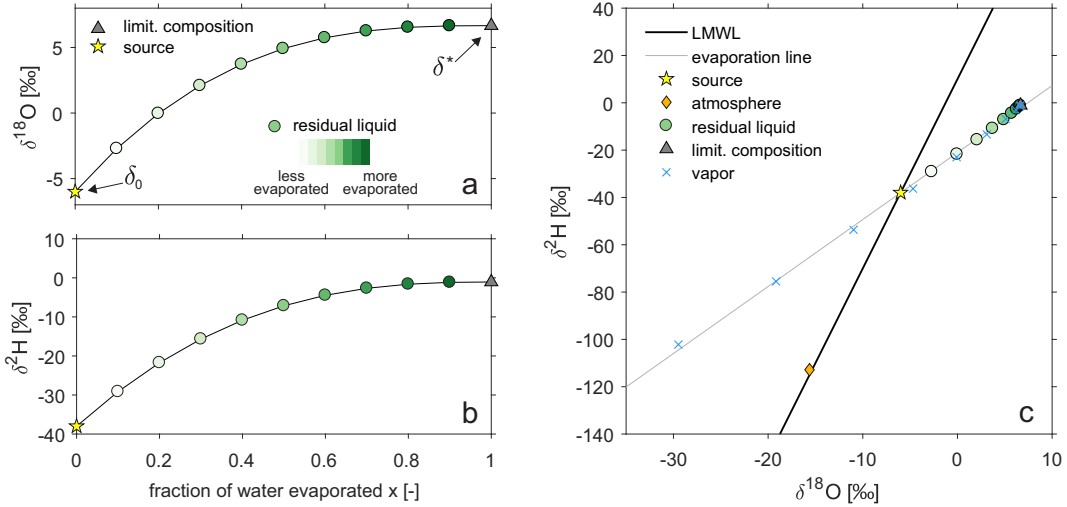

**Figure 2.** Introductory example showing the evolution of the isotopic composition of residual water $\delta_L$ for the case of an isolated volume of precipitation that evaporates into the atmosphere. The initial composition (source water) $\delta_0 = \delta_P$ is -6‰ (for $\delta^{18}$O) and -38‰ (for $\delta^2$H). Panels (a) and (b) show the oxygen and hydrogen isotopic composition for increasing fractions of evaporation (decreasing fraction of residual liquid) as they approach the limiting composition, while panel (c) shows the same isotope effects in a dual isotope plot.

## 2.2 Accounting for the seasonality of atmospheric variables

The degree of evaporative fractionation will vary seasonally, reflecting seasonal changes in temperature and relative humidity. The isotopic composition of precipitation will also vary seasonally, reflecting seasonal shifts in moisture sources, air mass trajectories, and cloud processes (Craig and Gordon, 1965; Rozanski et al., 1993). With this in mind, we explore how these two

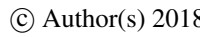



seasonal patterns jointly shape the isotopic composition of the residual liquid remaining after rainfall partly evaporates from a soil.

We consider a 12-month period and for each month we use the mean isotopic composition of precipitation as source water for the model outlined above. Each month's precipitation then undergoes a seasonally-varying amount of evaporation, and the isotopic composition of the residual water is determined separately for each month using equations (2-8), along with that month's average temperature and relative humidity. In this approach, the isotopic composition of monthly residual water is the isotopic composition that soil water would have if it were only influenced by the dynamics of precipitation and evaporation during the same month. This simplified approach does not explicitly account for in-soil mixing processes, whose effects are discussed in Section 3.

We apply this approach to real-world weather and precipitation data from the Vienna Hohe Warte station, Austria. The full isotopic dataset is freely available, along with temperature and vapour pressure data, from the Global Network of Isotopes in Precipitation (GNIP Database), provided by IAEA/WMO and accessible at: https://nucleus.iaea.org/wiser. We used the long-term mean monthly values of precipitation $\delta^{18}$O, air temperature, and vapour pressure, and then used these temperatures and vapour pressures to calculate relative humidity. Rather than using the measured precipitation $\delta^2$H values, we instead computed these using the equation for the LMWL at Hohe Warte: $\delta^2\mathrm{H} = 2.12 + 7.45\,\delta^{18}$O. This ensures that each precipitation sample plots exactly on the LMWL and thus aids visualization. The long-term mean monthly data is shown in Figure 3. All the timeseries exhibit pronounced seasonality. The seasonal temperature excursion is about 20°C, and monthly average $\delta^{18}$O ranges from -13‰ in winter to -6‰ in summer. The relative humidity ranges from roughly 0.85 in winter to 0.65 in spring and summer.

To investigate the effect of evaporation seasonality on residual liquid composition, we modeled the evaporation-to-precipitation fraction (the variable $x$) using sinusoidal cycles with different amplitudes and timing. We did not consider transpiration fluxes, since the isotopic effects of transpiration are generally considered to be negligible. Moreover, to keep the example simple, we did not consider the seasonality of precipitation flux, although this could be easily included. The parameter $n$ in equation (4) was fixed at 0.75 throughout the year.

## 3 Results

The isotopic compositions of different source waters (mean monthly values of precipitation from the Vienna Hohe Warte station) and of the residual liquid water after evaporation (computed through equations (2-8)) are shown in dual-isotope space in Figure 4. For this figure, we generated two hypothetical evaporation cycles, both peaking in July and having the same mean value $\bar{x}$ of 0.10 [-] but with different degrees of seasonality. The weakly seasonal cycle had a peak-to-peak amplitude of just 0.02, and the more strongly seasonal cycle had an amplitude of 0.16. These $x$ values are modest, representing conditions of limited evaporation as may be found in many temperate regions.

The source waters (shown as yellow stars) vary along the LMWL reflecting the seasonal variability of atmospheric moisture sources and conditions, with isotopically lighter precipitation during colder months. The simulated residual water samples



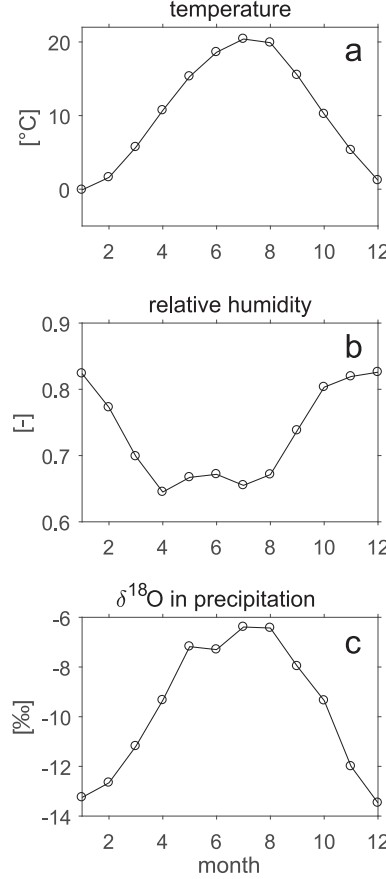

**Figure 3.** Long-term mean monthly air temperature (a), relative humidity (b) and oxygen isotopic composition in precipitation (c) for the station Vienna Hohe Warte, Austria

(shown as green dots) plot below the LMWL, with summer samples plotting farther from the LMWL than winter samples, reflecting their greater evaporative enrichment. The evaporation lines connecting individual source waters and residual waters are longer and shallower in summer than in winter, reflecting seasonal differences in temperature, relative humidity, and evaporated fraction $x$. As a result, the summer residual water samples plot farther away from the LMWL than the winter samples do, by an amount that reflects the seasonality in the evaporation process. The residual water samples follow a nearly linear trend (shown as a dashed line), which is markedly steeper than the evaporation lines for the individual source waters (shown as grey lines). The slopes of the evaporation lines range from 3.1 to 3.4; by contrast, the trendlines for the residual waters have slopes of 6.1 (Figure 4a) and 7.1 (Figure 4b), close to the assumed LMWL slope of 7.45. Note that whenever the residual water trendline has a slope that is close to that of the LMWL, the location of the intersection between these two lines will be highly uncertain.





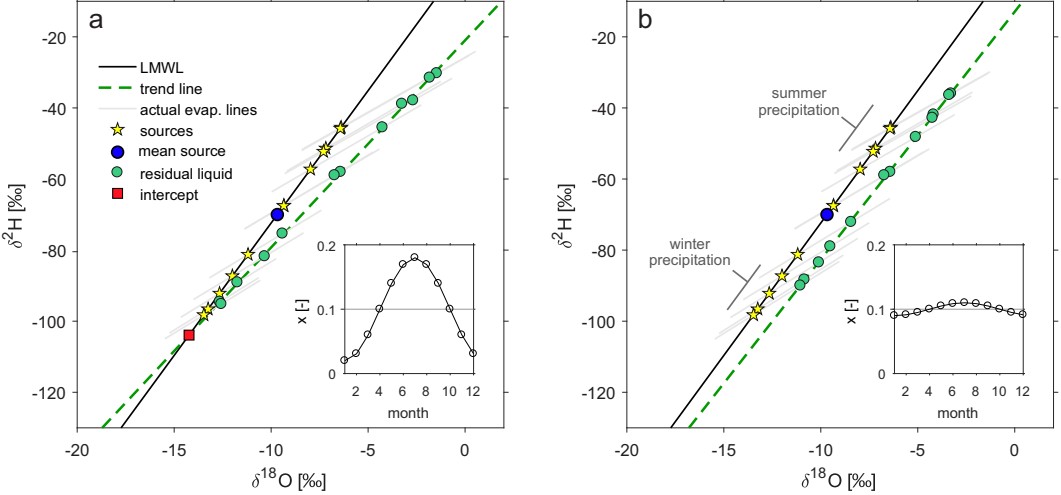

**Figure 4.** Effect of atmospheric seasonality on the isotopic composition of residual water from seasonally-varying precipitation. The evaporatively fractionated residual water samples (green dots) cluster around a trendline (dashed line) which is much steeper than the individual evaporation lines (grey lines). The effects of strong and weak evaporation seasonality are shown in panels (a) and (b), respectively. The insets show the assumed annual cycles in evaporated fractions $x$.

Because the simulated residual water samples can be fitted easily with a simple trendline, it may seem logical to interpret this trendline as an evaporation line, and to infer an apparent source water end-member from its intersection with the LMWL. In the case of an isolated water parcel that is progressively evaporated (as in Figure 2c), this approach could yield a reasonable estimate of the original source water. However, when residual water samples do not come from a single source (as in Figure 4), the trendline is not an evaporation line, and the intercept of this trend with the LMWL can lie far away from the average source water (and even far outside the range of all the source waters, as shown in Figure 4b).

Figure 5 illustrates how different degrees of seasonality in evaporation patterns may yield different trendlines in residual water samples, with different intercepts with the LMWL. The individual source waters and evaporation lines are the same as in Figure 4. Rather, the five trendlines in Figure 5 are associated with different seasonal evaporation cycles, which feature similar low evaporation fractions in winter ($x = 0.03 - 0.05$), but different evaporation fractions in summer (ranging from $x = 0.15$ to $x = 0.58$). The evaporation cycles with higher summer peaks correspond to trendlines with shallower slopes and less negative intersections with the LMWL. None of the intersections lie anywhere close to the true mean source water; indeed none lie within the range of the individual monthly source waters.

If the seasonal cycle of evaporative fractionation is not in phase with the seasonal cycle in source water composition (that is, if the most strongly fractionated sample is not also the one with the heaviest initial isotopic signature), the residual water samples will trace out a hysteresis loop. In Figure 6, the source waters are the same as those in Figure 4, but the seasonal evaporation cycle has been shifted by two months. The width of the resulting hysteresis loop depends on the amplitude of the





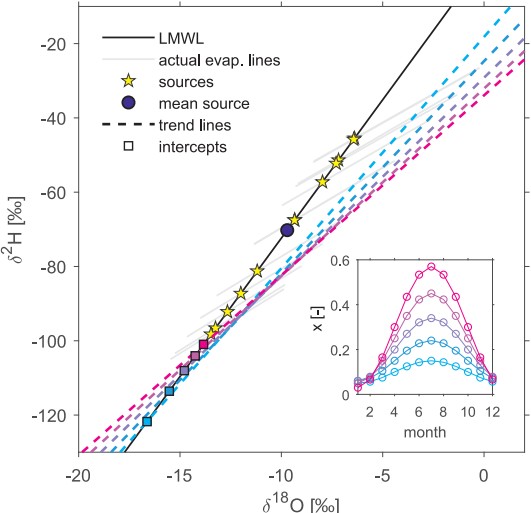

**Figure 5.** Examples of trendlines and intercepts arising from various seasonal evaporation patterns (inset).

seasonal cycle in evaporation, and how far out of phase it is with the seasonal cycle in precipitation isotopes. Even where such hysteresis loops exist in nature, they may be difficult to detect due to measurement uncertainties and environmental noise.

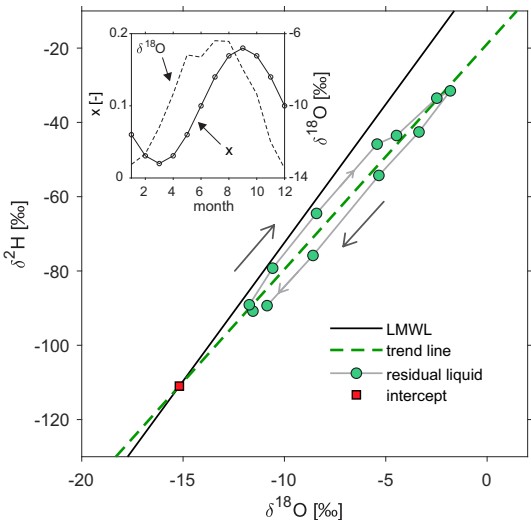

**Figure 6.** Hysteretic pattern arising in the computed residual liquid when the seasonality of source water composition and that of evaporation (inset) are shifted.

In Figures 4-6, each residual water sample is derived from a discrete monthly precipitation source water sample. Real-world soil waters, by contrast, can be expected to contain mixtures of waters with different ages, and thus different source water signatures and evaporative fractionation trajectories. For simplicity, we simulated the soil as a well-mixed reservoir that





integrates each month's residual waters. Mathematically this means that the composition of the soil pool is an exponentially weighted running average of the residual water samples shown in Figure 4a. For purposes of illustration we used a time constant of six months, such that the same-month contribution to each sample is roughly 15% and the contribution from the previous 12 months is roughly 86% of the total. The results are shown in Figure 7a.

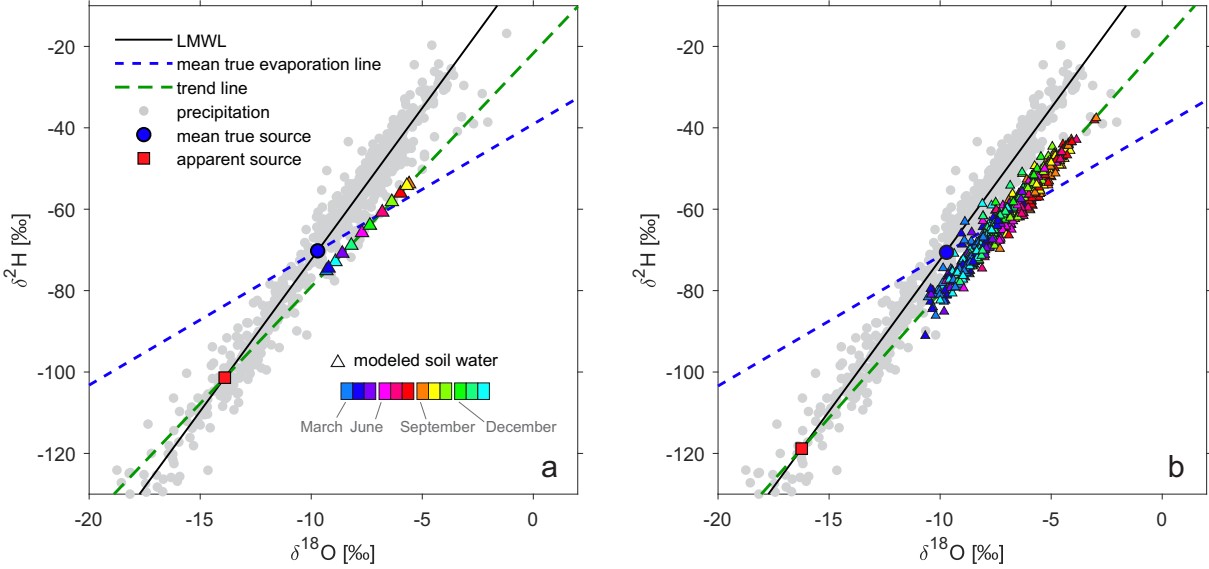

**Figure 7.** Isotopic composition of modeled soilwaters (triangles) obtained by mixing evaporated source waters through an exponential function. Soil water isotopic compositions are obtained starting from (a) long-term monthly sources (as in Figures 4-6), and (b) individual monthly isotopic sources, recorded at Vienna Hohe Warte station between 1961 and 2015 (grey dots).

The same procedure was used to create Figure 7b, except we considered as meteoric source waters each of the approximately 600 individual monthly $\delta^{18}$O and $\delta^2$H values available at Hohe Warte (the cloud of grey dots). These source waters were not constrained to lie along the LMWL, in contrast to the analyses presented above. These source waters were individually evaporated and fractionated, by amounts that depended on the individual monthly temperature and relative humidity (and the same seasonal cycle in the evaporated fraction $x$ that was assumed in Figures 4a, 6, and 7a). We then applied the same running

weighted time-averaging used in Figure 7a, with the resulting cloud of residual water samples shown in Figure 7b. The more that the residual water samples are time-averaged, the more their scatter will be compressed and the smaller the portion of the dual-isotope plot they will occupy, but their trendline will remain almost the same. The exponentially weighted averaging used here also introduces a time lag of roughly 3-4 months between the seasonal cycle in the source water and the seasonal cycle in the time-averaged soil water. For this reason, the isotopically heaviest soil water samples are found in October even

though the isotopically heaviest precipitation falls in the summer. (Different time constants in the weighted averaging would yield different lag intervals.)





Due to the scatter among the source water samples in Figure 7b, the evaporatively fractionated residual water samples are less collinear than in Figure 7a. Nonetheless, in both cases the trendlines intersect the LMWL far from the true mean source water. Because the intersection point lies within the range of the individual winter precipitation samples, however, there is a risk that one could incorrectly infer that it represented a winter-precipitation source water for the evaporated soil samples (when in fact the winter precipitation in these simulations has hardly been evaporated at all).

## 4 Discussion and Concluding Remarks

The analyses presented above serve as a reminder that ecohydrological isotope samples need to be understood as combining the effects of source variation, mixing, and fractionation. Indeed, in our examples, all three of these effects jointly determine the isotopic patterns in the evaporated soil samples.

All else equal, the greater the isotopic variability in precipitation (and thus the larger the range of source waters), the closer the slope of the evaporated samples will lie to the LMWL (Figure 8a). (Conversely, in the absence of any variability in precipitation, the evaporated samples would trace out evaporation lines instead.) All else equal, the greater the seasonality in evaporative fractionation, the more the slope of the evaporated samples will deviate from the LMWL (Figure 8b). The intercept of the trendline with the LMWL is driven purely by these geometric considerations, and has no significance in its own right.

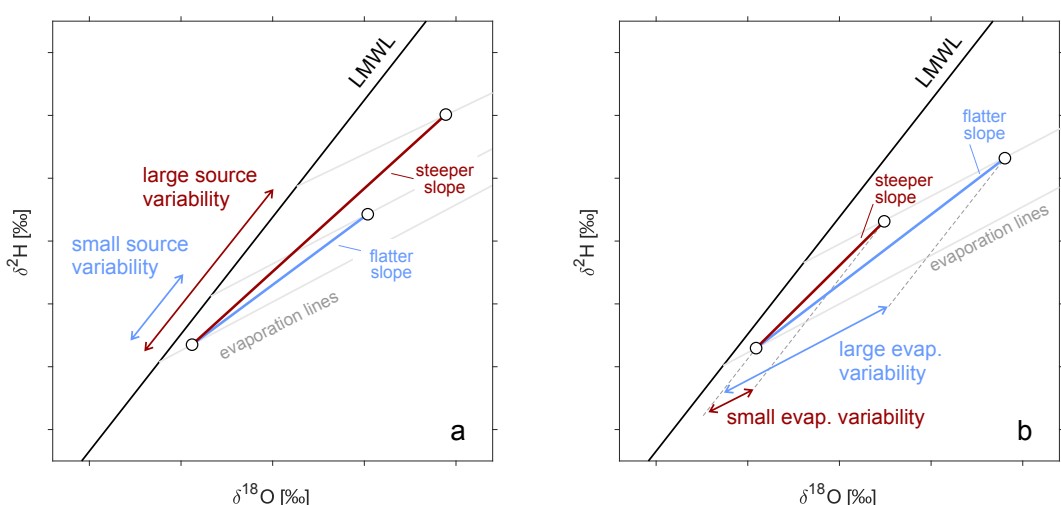

**Figure 8.** Effects of precipitation source variability (a) and evaporation variability (b) on the trendline that interpolates evaporated soil water samples (dots). The trendline is much steeper than the evaporation lines, unless there is no variability in the source water isotopic composition.

The seasonality of evaporative fractionation combines two factors: the variation in the slope of the evaporation line, and the variation in the amount of water lost to evaporation (as quantified by the evaporation fraction $x$), which determines how far out





on the evaporation line the evaporated samples are found. In most real-world situations, the second of these factors is likely to have a greater influence on the trendline of the evaporated samples (and thus on its intersection with the LMWL).

The more mixing the evaporated samples undergo, the more their variability will be compressed. And the more closely that the variations in precipitation isotopes and evaporation rates are synchronized, the more the evaporated samples will follow a

trendline; conversely, if they are out of phase, they will form a hysteresis loop.

Our analysis of the effects of variability in source signatures and evaporative fractionation has been couched in terms of seasonal patterns, but similar considerations apply to variations at other time scales as well. For example, under more arid conditions the evaporation line will have a flatter slope and evaporative losses will be greater, both factors that will push evaporated samples farther from the LMWL. If those atmospheric conditions are also correlated with isotopically heavier

source waters, the resulting residual water trendline will be similar to those we have simulated here (with a slope much steeper than a true evaporation line, and typically intersecting the LMWL far from the average source water).

Using the intersection between this trendline and the LMWL would give a heavily biased estimate of average source water, but what could give a better one? One can see from Figure 7 that a reasonable estimate of the average source water could be obtained by translating the individual evaporated samples back to the LMWL along assumed evaporation lines, yielding

estimates of their pre-evaporation compositions which are then averaged. Ideally the slope of each evaporation line would be determined from atmospheric conditions that are specific to each evaporated sample. But even where these are unknown, any reasonable estimate of the evaporation slope will yield much better results than the slope of the trendline through the evaporated samples.

We have chosen a relatively simple model to simulate the evaporative fractionation of the residual water samples. More

sophisticated models of evaporative fractionation in soil water have been proposed (see, e.g. Mathieu and Bariac, 1996; Soderberg et al., 2012; Dubbert et al., 2013; Good et al., 2014). Results from these models may provide more accurate estimates of the kinetic fractionation factors and thus of the slope of the evaporation line (and its variability). However, use of these models is unlikely to yield qualitatively different results from those in Figures 4-7, because other reasonable estimates of the fractionation factors (and thus of the slopes of the evaporation lines) will make little difference to the slope of the trendline

running through the evaporated samples. Our analysis also invokes the simplifying assumption that (for example) July's rainfall only evaporates under July conditions. But if some of July's rainfall is stored until August, September, October, etc. then some of it should also evaporate under those conditions. Arguably our analysis could be superseded by a detailed process model that simulates the time-dependent storage and release of water in soils. However, such a model would complicate the analysis considerably and we have no reason to believe that it would yield substantially different results.

The data and equations presented in this paper are not novel, and many readers will not be surprised by our conclusion that trendlines through evaporated samples can differ widely from true evaporation lines. Nonetheless, our analysis shows how residual water trendlines can result from the interplay of seasonally-varying isotopic inputs and evaporation rates, and shows that their intersection with the LMWL will generally be a highly unreliable guide to the average source water composition. Analyses that have used these trendlines to identify the compositions of source waters should therefore be re-examined.

Whether on seasonal or synoptic timescales, the regional and global energy dynamics that drive these variations in source



water composition and evaporative fractionation are likely to be widespread. Thus, although results for individual sites and time periods may differ in quantitative details from those presented here, we expect the qualitative patterns to be general.

*Code and data availability.* Isotopic data for the Vienna Hohe Warte station, Austria, are freely available from the Global Network of Isotopes in Precipitation (GNIP Database), provided by IAEA/WMO and accessible at: https://nucleus.iaea.org/wiser. A Matlab code to imple-
ment the equations described in Section 2 is provided as supplementary material.

*Competing interests.* TEXT

*Acknowledgements.* This article was inspired by discussions at the Workshop on "Isotope-based studies of water partitioning and plant-soil interactions in forested and agricultural environments", which was held in September 2017 in San Casciano in Val di Pesa, Florence, Italy. PB thanks the ENAC school at EPFL for financial support.



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
