# Peer review of "Effects of climatic seasonality on the isotopic composition of evaporating soil waters"

_Hydrology and Earth System Sciences, 2018_

## Referee Comment (RC1) · Anonymous Referee #1 · 16 Feb 2018

This article describes a modelling exercise to calculate the isotopic composition of evaporated soil water under seasonal varying isotopic composition of precipitation input and evaporation flux. In such the authors present a very insightful contribution to the difference between the real physical evaporation line and the trendline through the fractionated soil water compositions. The importance of this well structured and written paper is that it reminds the hydrological community on the physics of soil evaporation and shows that the intercept of this trendline on the LMWL is not the source of the soil water in real life hydrological cycle. This is only the case the evaporating source has a constant composition which is rarily the case in hydrological studies. However, the large seasonal variation in isotopic input and evaporation fluxes result in trendlines through soil water isotopic residuals having no information of the original source composition. The method and examples are elegantly simple and convincing. The paper has the correct length.

My only minor suggestions are that the title, intro and abstract could be formulated slightly stronger. Although I like the modesty with which the authors present it, they can be 'more firm' in the fact that soil water isotopic information has been used incorrectly. - For the title, maybe something more like: Soil isotopic composition through evaporation unveiled: the effect of etc.. Or: On the origin of soil water isotopic composition under seasonal isotopic input and evaporation

Second, I think the introduction (and the abstract equally) could be improved a little. It is a bit too 'modest'. For example: L9 (abstract) write "thus they are often not the evaporation line." Replace with: "we show these trendlines are not evaporation lines (under seasonal varying hydrological cycle)."

P2L13-P3L10: several time you write: "If...If....this should be valid if ....But what if the don't?" But if you know something is incorrect then also write it like this. For example: "The erroneous interpretation of these trendlines as single-source evaporation lines, ..."

P3L5: I suggest to the authors to rephrase this objective: In my opinion you do not test something, you use numerical experiments to shed insight in the origin of fractionated soil water samples.

Lastly, the authors mention xylem water samples but as they neglect transpiration in their analysis (I agree) I think it is more correct to only talk of soil water samples.

---

## Author Comment (AC1) · 13 Apr 2018

We thank Anonymous Referee #1 for their supportive comments on our manuscript. Referee #1 encouraged us to formulate several elements of the title, introduction, and abstract in stronger terms. We previously considered alternative titles that pointed more strongly to one of our main results, namely that so-called "evaporation lines" in soil waters are often artifacts of seasonality in precipitation isotopes and climate. We elected to use a more general title instead, because our analysis goes beyond just this point, to illuminate climatic effects on soil water isotopes more generally.

Likewise, the reviewer encouraged us to strengthen the main conclusions in our

abstract, "Here we use numerical experiments based on established isotope fractionation theory to show that these trendlines are often by-products of the seasonality in evaporative fractionation and in the isotopic composition of precipitation. Thus, they are often not true evaporation lines, and, if interpreted as such, can yield highly biased estimates of the isotopic composition of the source water". We believe it is important to keep the word "often" in both sentences, not as a matter of false modesty but instead as a matter of technical accuracy. Specifically, apparent "evaporation lines" may really be evaporation lines, in cases where the isotopic source of the evaporating soil water does not vary seasonally (for example, in locations where there is not significant variability in the isotopic composition of precipitation, or where soil water is supplied by exfiltration of old groundwater with constant isotopic composition). Thus our concluding statements would be inaccurate if they were stated as absolutes.

The reviewer notes, *"P2L13-P3L10: several time you write: "If...If....this should be valid if ....But what if the don't?" But if you know something is incorrect then also write it like this. For example: "The erroneous interpretation of these trendlines as single-source evaporation lines, ...""*. We do it this way because we are defining the key question to be explored in the paper. At this stage in the paper, we do not "know" (or at least we haven't proven) that interpreting trendlines as evaporation lines is erroneous, so we need to state it as a question rather than a conclusion. In revising the conclusions section of the manuscript, however, we will look for opportunities to state our conclusions even more clearly.

The reviewer's last comment is, *"Lastly, the authors mention xylem water samples but as they neglect transpiration in their analysis (I agree) I think it is more correct to only talk of soil water samples"*. We appreciate the point that we mention xylem in the abstract and introduction, but then we don't discuss it further. Rather than omitting xylem water entirely, we think that a better approach is to revise the paper to explicitly state

that because plant uptake is generally not strongly fractionating, xylem water composition will closely follow the composition of the soil water, and thus our conclusions will also apply to xylem water. We think that this is important because trendlines have been mis-interpreted as evaporation lines in both xylem water and soil water.

———————————————

---

## Referee Comment (RC2) · Anonymous Referee #2 · 16 Apr 2018

This is a very interesting, well written and important manuscript. It is also very timely, as more and more ecohydrological studies are using isotopes to determine the source of water that is taken up by plants and these researchers may be tempted to use the intercept of the trendline through the soil or xylem samples to infer the isotopic composition of the source water. This study shows that this is clearly wrong (except for locations where the isotopic composition of the source water does not vary seasonally) and fortunately also gives a suggestion on how to obtain a better estimate of the isotopic composition of the source water (P12L12-18). The beauty and impact of the manuscript lie in the simplicity of the approach that was followed and the very clear figures. I highly recommend rapid publication of this manuscript and have only very minor comments or suggestions.

[Figure]

1. P8L6: Isn't this the case for figure 4a as well?

2. P8L11: Can you clarify on what the x=0.58 is based (i.e. why did you choose this value and not another value)?

3. Perhaps add subheadings for the different results to make it even easier to follow or find the different analyses and results (from P6L26: evap from open water, from P8L14: asymmetric evap, P9L2: evap from soil water mixture).

4. Figure 6: I would find it useful if evaporation lines connecting the residual liquid samples and the source water were shown as well – like it is done in the other figures.

Very minor editorial suggestions:

Title: Replace 'soil waters' by 'soil water'

Abstract: Make the abstract more concrete by removing some of the "qualifiers": on P1L5: remove 'also', on P1L6: replace 'sometimes' by 'often'. Perhaps replace 'precipitation' by 'source water (and thus precipitation)'?

P1L17-P2L5: Either replace 'included' by 'focused on' or 'identifying' and 'quantifying' by 'the identification of' and 'quantification of'

P2L7: Remove 'at any location'?

P2L8: To avoid confusion with the trend line through samples of the remaining water, I would try to avoid using the word "trend" to describe the LMWL: replace 'follow a linear trend' by 'are linearly correlated' or 'plot on a linear line'

P2L12: Remove 'Collections of'?

P3L1-2: Replace 'waters' by 'water'

P3L15: Insert 'open water and' before 'soils' as it also describes the situations for shallow open water or small reservoirs

P3L12-14: Write in past tense 'simulate'-> 'simulated' and 'introduce' -> 'introduced'
P3L26: Add 'and' and 'It is….ratios' from the next sentence to the end of the sentence ending with 'equilibrium'

P3L26: Do you really need the 'super' here? Isn't it is just the ratio of the two isotopic ratios?

P4L16: Replace 'under' by 'for'?

P5L6: Move the part 'with parameters…n=1) to the caption of the figure. It is more informative there

P6L15: Replace 'ensures' by 'ensured'.

P6L16: Replace ', and thus aids visualization' by 'aiding interpretation and visualization'

P6L16-17L Remove 'data….All the' and add reference to Figure 3 at the end of the sentence (after 'seasonality')

P6L20: Replace 'evaporation seasonality' by 'seasonality in evaporation rates' to make it clearer that this is the rate or fraction of evaporation and not fractionation or the conditions during which evaporation takes place. Replace 'modeled' and 'using' by 'represented' and 'by'?

Caption Figure 4: Replace 'evaporation seasonality' by 'seasonality in evaporation rates (represented by x, the fraction of the initial volume that has evaporated)'

P6L9: Replace 'feature' by 'represent'

Caption figure 7: Replace 'are' by 'were'

P11L7: Remove 'ecohydrological' or put it in parentheses

P11L12: Replace 'out' by 'the'?

P12L12: Replace 'would give' by 'gives'

---

## Author Response (AR1)

**Response letter – manuscript hess-2018-40**

The comments provided by Editor and Reviewers have been reported below as italicized text. Our response follows point-by-point

**EDITOR**

> *Dear authors,*
>
> *thank you for your replies to the reviewer comments. As you have seen, both reviewers highly appreciate your manuscript. They do only suggest some very minor edits and recommend speedy publication after that. I agree with them. Therefore, please incorporate these points and then I will be glad to accept this very interesting and important contribution for publication in HESS.*
>
> *Best regards, Markus Hrachowitz*

We thank Dr. Hrachowitz for coordinating the review process. We indeed appreciated the comments provided by the referees. The manuscript has been modified to accommodate the reviewer's minor suggestions.

**ANONYMOUS REFEREE #1:**

> *This article describes a modelling exercise to calculate the isotopic composition of evaporated soil water under seasonal varying isotopic composition of precipitation input and evaporation flux. In such the authors present a very insightful contribution to the difference between the real physical evaporation line and the trendline through the fractionated soil water compositions. The importance of this well structured and written paper is that it reminds the hydrological community on the physics of soil evaporation and shows that the intercept of this trendline on the LMWL is not the source of the soil water in real life hydrological cycle. This is only the case the evaporating source has a constant composition which is rarily the case in hydrological studies. However, the large seasonal variation in isotopic input and evaporation fluxes result in trendlines through soil water isotopic residuals having no information of the original source com position. The method and examples are elegantly simple and convincing. The paper has the correct length.*

We thank Anonymous Referee #1 for their supportive comments on our manuscript.

> *My only minor suggestions are that the title, intro and abstract could be formulated slightly stronger. Although I like the modesty with which the authors present it, they can be 'more firm' in the fact that soil water isotopic information has been used incorrectly. - For the title, maybe something more like: Soil isotopic composition through evaporation unveiled: the effect of etc.. Or: On the origin of soil water isotopic composition under seasonal isotopic input and evaporation*

Referee #1 encouraged us to formulate several elements of the title, introduction, and abstract in stronger terms. We previously considered alternative titles that pointed more strongly to one of our main results, namely that so-called "evaporation lines" in soil waters are often artifacts of seasonality in precipitation

isotopes and climate. We elected to use a more general title instead, because our analysis goes beyond just this point, to illuminate climatic effects on soil water isotopes more generally. Following this reviewer's comment, however, we have modified the final paragraph of the paper to formulate our conclusions in stronger terms: "Analyses that have used these trendlines to identify the compositions of source waters may be substantially in error, and therefore should be re-examined."

> *Second, I think the introduction (and the abstract equally) could be improved a little. It is a bit too 'modest'. For example: L9 (abstract) write "thus they are often not the evaporation line." Replace with: "we show these trendlines are not evaporation lines (under seasonal varying hydrological cycle)."*

Similar to the point before, the reviewer encouraged us to strengthen the main conclusions in our abstract, "Here we use numerical experiments based on established isotope fractionation theory to show that these trendlines are often by-products of the seasonality in evaporative fractionation and in the isotopic composition of precipitation. Thus, they are often not true evaporation lines, and, if interpreted as such, can yield highly biased estimates of the isotopic composition of the source water." We believe it is important to keep the word "often" in both sentences, not as a matter of false modesty but instead as a matter of technical accuracy. Specifically, apparent "evaporation lines" may really be evaporation lines, in cases where the isotopic source of the evaporating soil water does not vary seasonally (for example, in locations where there is not significant variability in the isotopic composition of precipitation, or where soil water is supplied by exfiltration of old groundwater with constant isotopic composition). Thus our concluding statements would be inaccurate if they were stated as absolutes.

> *P2L13-P3L10: several time you write: "If...If....this should be valid if ....But what if the don't?" But if you know something is incorrect then also write it like this. For example: "The erroneous interpretation of these trendlines as single-source evaporation lines, ..."*

The use of hypothetical sentences starting with "if…" is needed here because we are defining the key question to be explored in the paper. At this stage in the paper, we do not "know" (or at least we haven't proven) that interpreting trendlines as evaporation lines is erroneous, so we need to state it as a question rather than a conclusion. In any case, as noted in a previous comment, we have modified the final paragraph of the paper to formulate our conclusions in stronger terms.

> *P3L5: I suggest to the authors to rephrase this objective: In my opinion you do not test something, you use numerical experiments to shed insight in the origin of fractionated soil water samples.*

We thank Referee #1 for this suggestion. However, at P3L5 we simply state that "To date, no benchmark experiment has tested whether, and under what conditions, the trendline passing through fractionated soil water samples correctly identifies their source water". That is, this is a comment on the current state of the science, and we believe that it is factually correct. Our objectives are already outlined in the following sentences (P3 L7-10). In any case, our analysis can be viewed as a falsification test for the widespread practice of inferring source water compositions from such trendlines, because we show that it yields the wrong answer, even in a simple model framework that is free of many of the complicating factors that would be present in any real-world case.

> *Lastly, the authors mention xylem water samples but as they neglect transpiration in their analysis (I agree) I think it is more correct to only talk of soil water samples.*

We appreciate the point that we mention xylem in the abstract and introduction, but then we don't discuss it further. Rather than omitting xylem water entirely, we thought that a better approach was to revise the paper to explicitly state that: "Because plant uptake is generally not strongly fractionating, isotopic variations in soil water are likely to be transferred to plant xylem, and thus we expect that our conclusions will also apply to xylem water as well." We think that this is important because trendlines have been mis-interpreted as evaporation lines in both xylem water and soil water.

**ANONYMOUS REFEREE #2:**

> *This is a very interesting, well written and important manuscript. It is also very timely, as more and more ecohydrological studies are using isotopes to determine the source of water that is taken up by plants and these researchers may be tempted to use the intercept of the trendline through the soil or xylem samples to infer the isotopic composition of the source water. This study shows that this is clearly wrong (except for locations where the isotopic composition of the source water does not vary seasonally) and fortunately also gives a suggestion on how to obtain a better estimate of the isotopic composition of the source water (P12L12-18). The beauty and impact of the manuscript lie in the simplicity of the approach that was followed and the very clear figures. I highly recommend rapid publication of this manuscript and have only very minor comments or suggestions.*

We thank Reviewer #2 for these very positive comments on the manuscript

> *1. P8L6: Isn't this the case for figure 4a as well?*

The sentence at Page 8 Lines 4-6 reads: "However, when residual water samples do not come from a single source (as in Figure 4), the trendline is not an evaporation line, and the intercept of this trend with the LMWL can lie far away from the average source water (and even far outside the range of all the source waters, as shown in Figure 4b)". We believe that only in Figure 4b does the intercept lies "far outside the range of all the source waters", as in Figure 4a it is rather close to the winter sources. The sentence has now been reformulated as: "However, when residual water samples do not come from a single source, the trendline is not an evaporation line, and the intercept of this trend with the LMWL can lie far away from the average source water (Fig. 4a); the intercept can even lie far outside the range of all the source waters (Fig. 4b)".

> *2. P8L11: Can you clarify on what the x=0.58 is based (i.e. why did you choose this value and not another value)?*

The seasonal patterns shown in Figure 5 were generated using sinusoids with growing means and growing amplitudes that are a function of the mean. Using mean = 0.3 and amplitude = 0.9*mean = 0.27, one obtains a maximum value x = 0.57 (and not 0.58, we see that the manuscript was slightly imprecise). To reduce the emphasis on this detail, we have used the expression "…which feature similar low evaporation fractions in winter (roughly x=0.04), but different evaporation fractions in summer (roughly x=0.15 to x=0.60)."

*3. Perhaps add subheadings for the different results to make it even easier to follow or find the different analyses and results (from P6L26: evap from open water, from P8L14: asymmetric evap, P9L2: evap from soil water mixture).*

Thanks for the suggestion. We have separate the Results section into two subsections: 3.1 Seasonal patterns in evaporated soil waters; 3.2 Mixtures of evaporated soil waters

*4. Figure 6: I would find it useful if evaporation lines connecting the residual liquid samples and the source water were shown as well – like it is done in the other figures.*

Thanks for the suggestion. We have modified Figure 6 to include the actual evaporation lines and the source waters.

*Very minor editorial suggestions:*

*Title: Replace 'soil waters' by 'soil water'*
*Abstract: Make the abstract more concrete by removing some of the "qualifiers": on P1L5: remove 'also', on P1L6: replace 'sometimes' by 'often'. Perhaps replace 'precipitation' by 'source water (and thus precipitation)'?*
*P1L17-P2L5: Either replace 'included' by 'focused on' or 'identifying' and 'quantifying' by 'the identification of' and 'quantification of'*
*P2L7: Remove 'at any location'?*
*P2L8: To avoid confusion with the trend line through samples of the remaining water, I would try to avoid using the word "trend" to describe the LMWL: replace 'follow a linear trend' by 'are linearly correlated' or 'plot on a linear line'*
*P2L12: Remove 'Collections of'?*
*P3L1-2: Replace 'waters' by 'water'*
*P3L15: Insert 'open water and' before 'soils' as it also describes the situations for shallow open water or small reservoirs*
*P3L12-14: Write in past tense 'simulate'-> 'simulated' and 'introduce' -> 'introduced'*
*P3L26: Add 'and' and 'It is. . . .ratios' from the next sentence to the end of the sentence ending with 'equilibrium'*
*P3L26: Do you really need the 'super' here? Isn't it is just the ratio of the two isotopic ratios?*
*P4L16: Replace 'under' by 'for'?*
*P5L6: Move the part 'with parameters. . .n=1) to the caption of the figure. It is more informative there*
*P6L15: Replace 'ensures' by 'ensured'.*
*P6L16: Replace ', and thus aids visualization' by 'aiding interpretation and visualization'*
*P6L16-17L Remove 'data. . ..All the' and add reference to Figure 3 at the end of the sentence (after 'seasonality')*
*P6L20: Replace 'evaporation seasonality' by 'seasonality in evaporation rates' to make it clearer that this is the rate or fraction of evaporation and not fractionation or the conditions during which evaporation takes place. Replace 'modeled' and 'using' by 'represented' and 'by'?*
*Caption Figure 4: Replace 'evaporation seasonality' by 'seasonality in evaporation rates (represented by x, the fraction of the initial volume that has evaporated)'*
*P6L9: Replace 'feature' by 'represent'*
*Caption figure 7: Replace 'are' by 'were'*

*P11L7: Remove 'ecohydrological' or put it in parentheses*
*P11L12: Replace 'out' by 'the'?*
*P12L12: Replace 'would give' by 'gives'*

Thanks for these suggestions. We have used the majority of the suggested minor edits in the revised manuscript.

[revised manuscript text omitted]